# Micro-Computed Tomography Analysis of Subchondral Bone Regeneration Using Osteochondral Scaffolds in an Ovine Condyle Model

Taylor Flaherty, Maryam Tamaddon and Chaozong Liu *

Institute of Orthopaedic & Musculoskeletal Science, University College London, Royal National Orthopaedic Hospital, Stanmore HA7 4LP, UK; tcrntmf@ucl.ac.uk (T.F.); m.tamaddon@ucl.ac.uk (M.T.)
* Correspondence: chaozong.liu@ucl.ac.uk

**Abstract:** Osteochondral scaffold technology has emerged as a promising therapy for repairing osteochondral defects. Recent research suggests that seeding osteochondral scaffolds with bone marrow concentrate (BMC) may enhance tissue regeneration. To examine this hypothesis, this study examined subchondral bone regeneration in scaffolds with and without BMC. Ovine stifle condyle models were used for the in vivo study. Two scaffold systems (8 mm diameter and 10 mm thick) with and without BMC were implanted into the femoral condyle, and the tissues were retrieved after six months. The retrieved femoral condyles (with scaffold in) were examined using micro-computed tomography scans (micro-CT), and the micro-CT data were further analysed by ImageJ with respect to trabecular thickness, bone volume to total volume ratio (BV/TV) ratio, and degree of anisotropy of bone. Statistical analysis compared bone regeneration between scaffold groups and sub-set regions. These results were mostly insignificant ($p < 0.05$), with the exception of bone volume to total volume ratio when comparing scaffold composition and sub-set region. Additional trends in the data were observed. These results suggest that the scaffold composition and addition of BMC did not significantly affect bone regeneration in osteochondral defects after six months. However, this research provides data which may guide the development of future treatments.

**Keywords:** bone regeneration; scaffold; tissue engineering; micro-computed tomography; regenerative medicine; bone tissue–material interaction

## 1. Introduction

The osteochondral unit is a composite system of articular cartilage, calcified cartilage, and subchondral bone within synovial joints. Degenerative and traumatic injuries within this unit can damage tissue growth and integrity, leading to a general loss of stability and functionality within the entire joint [1–4]. The commonality of such injuries has led to the development of a number of surgical treatments. Unfortunately, research has shown that these treatments are associated with persistent joint pain, limited mobility, further degradation of the joint, and an overall worsened quality of life [2,5,6]. Osteochondral tissue engineering (OCTE), a technique which utilises a combination of tissue biology, material science, bioengineering, and cell transplantation to aid osteochondral tissue regeneration, was developed to combat the complications seen with previous treatments [7–9].

Biomimetic scaffolds utilised within OCTE mimic the surrounding tissues to act as a template for cell adhesion and promote adequate tissue integration and regeneration. These scaffolds may be composed of any number of biomaterials, including metals such as titanium, natural polymers such as collagen, synthetic polymers such as polyethylene, and ceramics [9–12]. Stem cells are often employed in conjunction with scaffolds; however, this is an expensive, multi-step procedure with a high risk of donor site morbidity in both autogenic and allogenic donors [6,13,14]. These limitations have led researchers to consider seeding scaffolds with bone marrow concentrate (BMC) rather than stem cells [14,15].

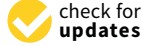



Studies suggest that the use of BMC increases the quantity and quality of tissues within the osteochondral unit because it promotes both osteogenesis and chondrogenesis. It has also been suggested that BMC enhances joint health and longevity because it contains an amalgamation of allogenic platelets, progenitor cells, lymphocytes, and growth factions in addition to stem cells. This method also eliminates the timeliness and high cost affiliated with the sole use of stem cells [16–18].

It is important to study tissue regeneration in OCTE treatments in order to assess success, efficacy, and safety. Methods commonly applied to study regenerated tissues (e.g., histology) are destructive, thus limiting the information extracted from the samples. Imaging methods such as micro-computed tomography (micro-CT) may be used to address this issue. Micro-CT machines scan specimens to create 2D pixel maps in thin slices through an entire object; these can then be used to generate 3D models for digital data collection [19]. This is an attractive method of analysis because it is non-destructive, detailed, and efficient. Research has also shown that micro-CT data can be successfully utilised for analysing bone and cartilage structure, the quantity of tissue growth, and scaffold structure whilst also rendering results consistent with traditional techniques [19–22].

In this study, a novel osteochondral scaffold based on a composite matrix of titanium, polylactic acid (PLA), and collagen-polylactic co glycolic acid (PLGA) was developed. The scaffolds were evaluated using ovine stifle condyle models. The bone formation within the bone compartment of the scaffolds was examined using micro-CT. The present study analysed the performance of this scaffold with and without BMC cells to determine if the inclusion of BMC contributed to the significant increase in bone regeneration. Subchondral bone was chosen as the focal tissue because it supports the articular cartilage, endures the biomechanical forces within the joint, and is the main hub for active tissue remodelling within the osteochondral unit. Micro-CT images were further analysed using ImageJ to determine trabecular thickness (TbTh), bone volume to total volume ratio (BV/TV), and the degree of anisotropy (DA) within the tested samples, offering insight into the regeneration of the subchondral bone.

## 2. Materials and Methods

### 2.1. Scaffolds Tested

Two osteochondral systems were used in this study—titanium matrix reinforced collagen tri-layered composite osteochondral scaffold (Ti–collagen) and collagen-hydroxyapatite tri-layered osteochondral scaffold (HA–collagen). The lower layer of the Ti–Collagen scaffold corresponding to bone tissue was designed as a porous Ti matrix to allow for bone ingrowth and vascularisation. Pore sizes of 300–800 μm are considered beneficial for bone ingrowth [23]. As such, the Ti matrix was designed with a strut diameter of 0.5 mm and a pitch size of 0.5 mm. The Ti was manufactured from Ti6Al4V alloy using a Laser Sintering (LS) system for metal (Lincotek, Trento, Italy) in compliance with ASTM F2924, as previously described [12]. The description of the osteochondral scaffolds has been reported elsewhere [24]. Four test groups were arranged within this research; Group 1: HA–collagen scaffold; Group 2: HA–collagen scaffold with BMC; Group 3: Ti–PLA–collagen scaffold; and Group 4: Ti–PLA–collagen scaffold with BMC. The test group arrangement is summarised in Table 1. The HA–collagen scaffolds contained a lower layer of 30% collagen and 70% hydroxyapatite, a middle layer of 60% collagen and 40% hydroxyapatite, and a top layer of 100% collagen, as schemed in Figure 1a. The preparation of these scaffolds was outlined in [25]. Alternatively, Ti–PLA–collagen/PLGA scaffolds contained a lower layer composed of a porous titanium lattice generated with direct metal laser sintering, a middle layer of polylactic acid generated with fuse deposition printing, and an uppermost layer of collagen/poly(lactic-co-glycolic acid) generated with casting and freeze-drying, as shown in Figure 1b. The biomaterials were selected because research has shown that metals are biocompatible, chemically stable, and easily manufactured to provide strong mechanical support to the overlying collagen layer [12,26], while polylactic acid middle layer joins the collagen layer with the Ti-matrix together to form a tri-layered osteochondral

scaffold [27–29]. Scaffolds which included BMC generated from the test subjects were submerged in 1.0 mL of solution for 20 min prior to implantation.

**Table 1.** Scaffold group, composition, and sample size. One sample in Group 3 was lost during data acquisition.

| Group | Scaffold | Group Size |
|---|---|---|
| 1 | HA–collagen | 6 |
| 2 | HA–collagen + BMC | 6 |
| 3 | Ti–PLA–collagen/PLGA | 5 |
| 4 | Ti–PLA–collagen/PLGA + BMC | 6 |

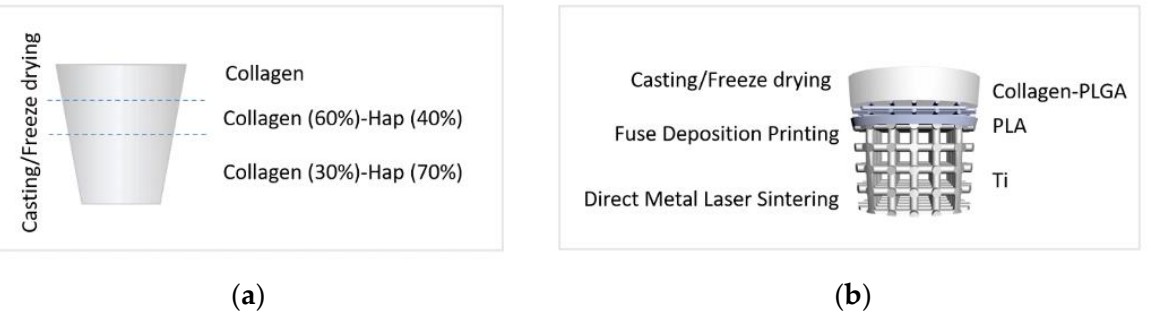

(**a**)

(**b**)

**Figure 1.** Composition of the tested scaffolds. (**a**) Collagen/hydroxyapatite scaffold; and (**b**) Ti–PLA–collagen/PLGA scaffold.

### 2.2. Surgical Procedure

Adult female sheep (Mules, Royal Veterinary College, London, UK) were used in this study. The in vivo animal study was performed in compliance with the United Kingdom Home Office Animals (Scientific Procedures) Act of 1986 because large animal models offer data which are more physiologically similar to humans than small animal models such as mice or rabbits [30]. After receiving general anaesthesia, bone marrow aspirates were taken from the posterior iliac crest, processed with NTL Biologica kit (Oxfordshire, UK), and 1.0 mL of the bone marrow concentrate was used to submerge the scaffolds. Osteochondral defects 8.0 mm in diameter and 10.0 mm in depth were then created in the animal's medial femoral condyles using surgical drills and reamers. Then, the scaffolds were press fit into the defects. Each sheep also had fentanyl patches on pre-operatively until day 3. Animals were housed in individual pens for four days post-surgery and then transferred to group pens for the remainder of the study. Post euthanasia, the joints were opened and the defect site and surrounding joint tissues were examined. No skin reaction or inflammation was observed. Tissue reactions ranged from minimal to moderate within the groups. Test subjects were sacrificed at six months post-operation, after which time the femoral condyles were excised and analysed.

### 2.3. Micro-CT Examination and Image Analysis

Micro-CT scan was performed using a Skyscanner 1172 (Bruker Kontich, Belgium) with kV X-ray source, 100 mA (pixel size 16.89 μm), and an aluminium + copper filter. Data were then reconstructed with NRecon software (Version: 1.7.1.6) and a circular region of interest 10 mm in diameter was applied to all micro-CT slices using CTAn software (Version: 1.17.7.2, Bruker) to include the original 8 mm defect region and an additional 2 mm of bone layer surrounding the scaffold and defect.

Each of the 23 scaffold samples contained between 500 and 700 individual micro-CT slices. To accommodate for limitations in processing large amounts of data, all slices for each scaffold sample were divided into three equal sub-sets, where Sub-set 1 was always closest to the knee joint and Sub-set 3 was always the furthest from the knee joint

(Figure 2). All micro-CT slices per sub-set were first imported into ImageJ. These were then stacked into a single file and converted to 8-bit binary images (Figure 3a) using threshold adjustments; control samples could be converted automatically whilst Ti–PLA–collagen/PLGA samples required manual conversion to ensure that ImageJ did not mistake the titanium for bone tissue. BoneJ, an ImageJ plugin developed specifically for quantified data extraction from microscopic and macroscopic skeletal samples [31,32], was then used to collect TbTh, BV/TV, and DA data.

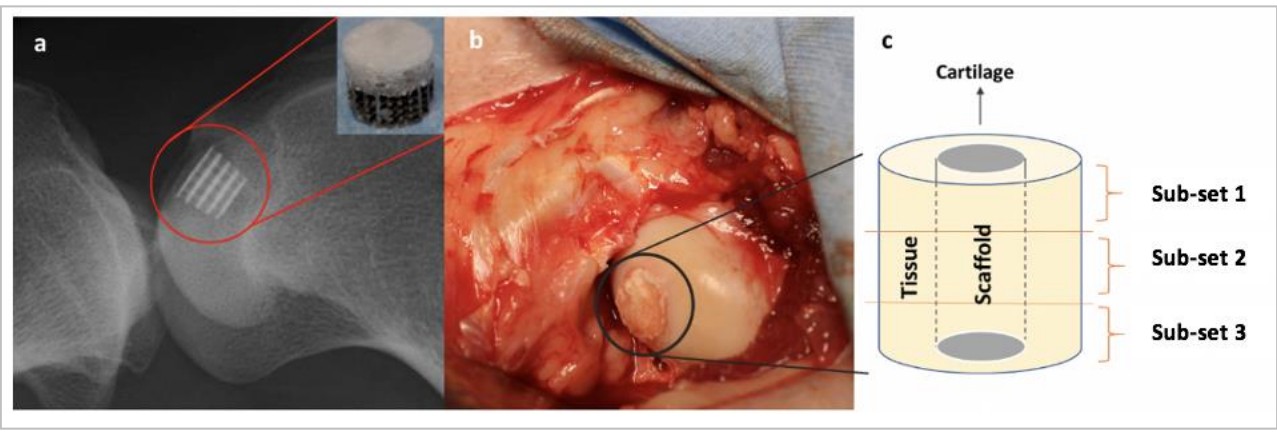

**Figure 2.** Schematic showing scaffold and sub-set placement. (**a**) X-ray showing the anatomical placement of a Ti–PLA–collagen/PLGA scaffold; (**b**) scaffold implanted in the condyle during surgery; and (**c**) subchondral bone surrounding the scaffold was divided into three sub-sets for microstructural analysis. Sub-set 1 is the most inferior third, Sub-set 2 is the intermediate third, and Sub-set 3 is the most superior third.

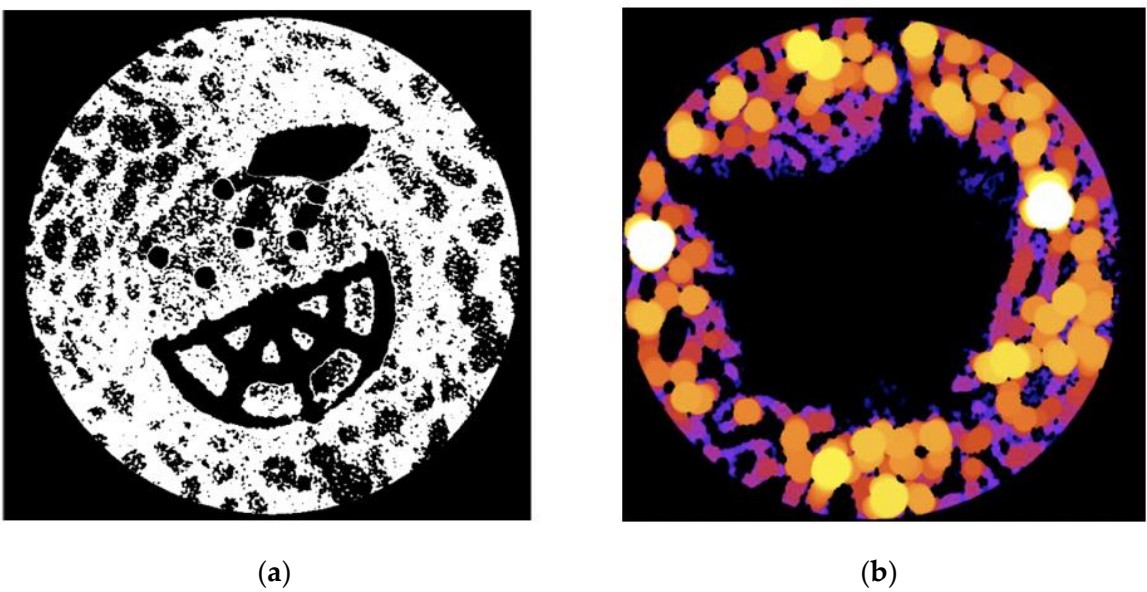

(**a**)      (**b**)

**Figure 3.** (**a**) Example of an 8-bit binary image from a Ti–PLA–collagen/PLGA sample. White = bone tissue; and (**b**) example of a trabecular thickness (TbTh) heatmap from a control sample. Black = no tissue or scaffold. Dark colours (purple, blue) = thinner tissue. Bright colours (orange, yellow) = thicker tissue.

Trabecular thickness quantifies trabecular bone growth by defining the thickness as the diameter of the largest sphere that can fit within the tested structure [32,33]. When selecting 'Thickness' in BoneJ, the TbTh mean and standard deviation are provided for the sample along with heatmaps to visualise areas of greater or lesser tissue growth (Figure 3b). The bone volume to total volume ratio (BV/TV) measurement provides the volume of bone per unit in order to show how much growth has occurred [29,32]. When utilising this feature,

a BV/TV ratio in the form of a percentage is provided. Finally, the degree of anisotropy (DA) was assessed to provide insight into the orientation and biomechanical durability of newly deposited tissue [32,34]. Standard values for directions (2000), lines per direction (10,000), and sampling increment (1.73) which are built within BoneJ were used to generate the DA. The value provided is always between 0 and 1, where larger values indicate more homogenous bone deposition and thus increased biomechanical stability [34,35].

### 2.4. Macroscopic Evaluation of Tissue Regeneration

All samples were sectioned to evaluate tissue regeneration macroscopically and observe scaffold degradation following the six month implantation period. The condyles were first dehydrated through a series of alcohol and transferred to LR White Resin (London Resin Company, London, UK). The resin was set using an accelerator according to the manufacturer's recommendation. Undecalcified 300 μm thick sections along the long axis of all condyles were cut in a parallel direction using a diamond saw micro sectioning system (Exakt Apparatebau, Norderstedt, Germany) and were imaged.

### 2.5. Statistical Analysis

Using Microsoft Excel, the total average TbTh, BV/TV, and DA were calculated from the three sub-sets for each tested scaffold. All sub-sets and total average data per sample were then inputted into IBM Statistical Package for Social Sciences (SPSS) for distribution of data analysis. Shapiro–Wilks tests and quantile–quantile plots (Q–Q plots) determined that none of the data significantly differed from normality, thus determining that parametric assessment could be utilised. Three separate two-way analysis of variance (ANOVA) tests were run to assess the impacts that scaffold composition, the addition of BMC, and sub-set region had on each of the three regeneration measurements. ANOVA1 simply compared the measurements for each of the four scaffold groups. ANOVA2 analysed the measurements for the two scaffold compositions and those with/without BMC. ANOVA3 compared both scaffold composition and addition of BMC between the three sub-set regions. A *p*-value of less than 0.05 was considered significant.

## 3. Results

### 3.1. Trabecular Thickness Varies Amongst Tested Scaffolds

Trabecular thicknesses within the examined region of interest for all samples were calculated; these results are summarised in Table 2. It was observed that the trabecular thickness for all test groups had close values ranging from 0.26 mm to 0.33 mm. Scaffold Group 2 demonstrated the largest overall average of 0.30 mm, followed by scaffold Group 1 and 3 with 0.29 mm and Group 4 with 0.28 mm. Within Sub-set 1, TbTh measurements ranged from 0.18 mm to 0.43 mm. Measurements in Sub-set 2 ranged from 0.17 mm to 0.38 mm, while in Sub-set 3, TbTh measurements ranged from 0.15 mm to 0.47 mm (Figure 4). The average TbTh measurement across all groups was 0.29 mm.

**Table 2.** Average scaffold and sub-set measurements for trabecular thickness.

|               | Group 1  | Group 2  | Group 3  | Group 4  |
| ------------- | -------- | -------- | -------- | -------- |
| Sub-Set 1     | 0.30 mm  | 0.29 mm  | 0.26 mm  | 0.31 mm  |
| Sub-Set 2     | 0.28 mm  | 0.29 mm  | 0.28 mm  | 0.27 mm  |
| Sub-Set 3     | 0.31 mm  | 0.33 mm  | 0.31 mm  | 0.27 mm  |
| Total Average | 0.29 mm  | 0.30 mm  | 0.29 mm  | 0.28 mm  |

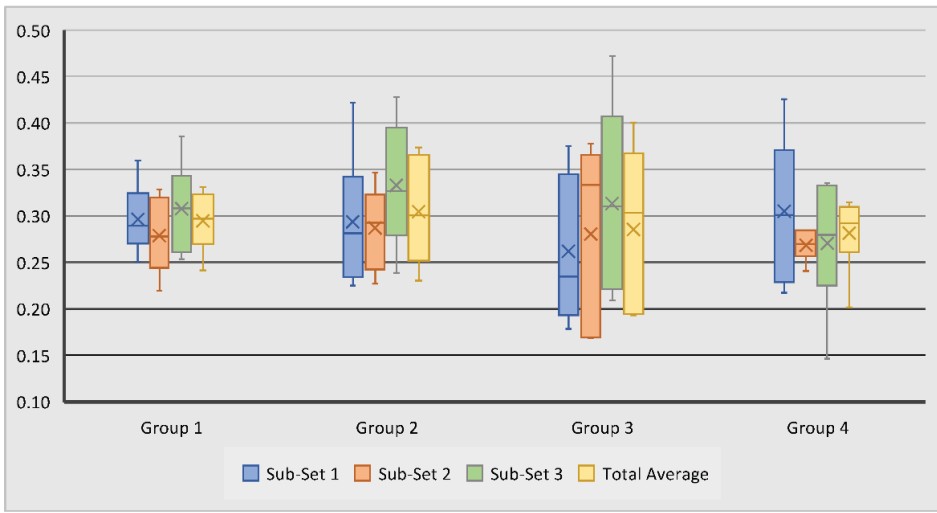

**Figure 4.** Variation of subchondral bone trabecular thickness in tested groups and sub-sets. No statistically significant differences were observed between the measurements.

In Group 1 and 2, the highest average TbTh was observed in Sub-set 3, suggesting that the region of the defect furthest from the joint surface experienced the greatest trabecular thickness. This was surprising because this was not visually apparent on many of the heat maps from these samples (Figure 5). This may have resulted from Sub-set 3 being furthest from the joint and thus experiencing fewer biomechanical stressors. However, Sub-set 1 within these samples did not consistently experience the least TbTh, suggesting that biomechanical stressors may not be the sole cause for the results observed within these scaffold groups.

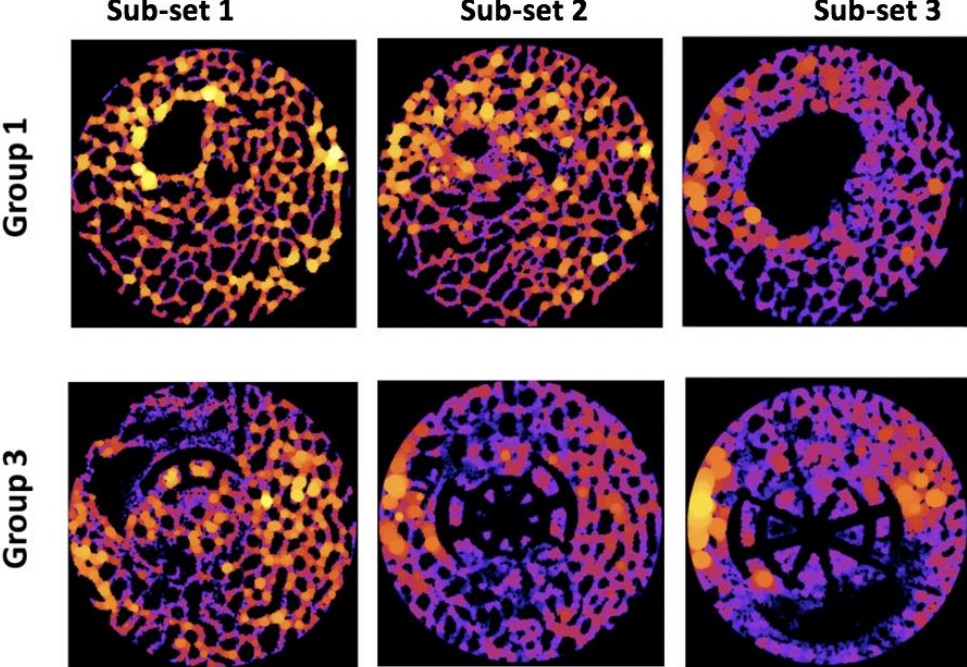

**Figure 5.** Examples of trabecular thickness heat maps. Black = no tissue or scaffold. Dark colours (purple, blue) = thinner tissue. Bright colours (orange, yellow) = thicker tissue.

Similar to Group 1 and 2, Sub-set 3 also experienced the greatest TbTh average in scaffold Group 3; this was again surprising because the visual assessment of the heat maps

did not overwhelmingly indicate this (Figure 5). Within this sample, sub-set 1 contained the lowest average measurement and Sub-set 2 the median measurement, indicating that this scaffold consistently produced greater tissue thickness in regions further from the articular surface of the joint.

Unlike with the other groups, the average TbTh within scaffold Group 4 was largest in Sub-set 1; this was visually apparent on some, though not all, of the heat maps. This trend change within this measurement suggests that this scaffold combined with the BMC was the only sample to generate thicker bone tissue nearest to the joint after six months.

All results for this measurement were statistically insignificant ($p > 0.05$). This suggests that, on average, the novel scaffold and addition of BMC to scaffolds did not significantly affect bone tissue thickness after six months of scaffold implantation. Other notable trends within the data were observed, however. For example, control scaffolds with BMC exhibited a slightly higher average TbTh than control scaffolds without BMC, but novel scaffolds without BMC exhibited a slightly higher average value than those with BMC. This suggests that the addition of BMC did not consistently promote greater TbTh measurements, but the interaction of BMC with its substrate may be an important factor in bone regeneration.

### 3.2. Bone Volume to Total Volume Ratio Varies across Samples

Bone volume to total volume ratio measurements were used to examine regenerated bone volume across samples. Scaffold Group 2 experienced the largest average BV/TV measurement of 39.68%, followed by Group 3 with 36.32%, Group 1 with 36.27%, and Group 4 with 32.8%. Within Sub-set 1, measurements ranged from 23.3% to 49.1%, with Group 4 showing the highest average BV/TV at 40.15%. Measurements in Sub-set 2 ranged from 15.7% to 50.2%, and Group 2 showed the highest average measurements at 39.2%. Sub-set 3 BV/TV measurements ranged from 14.3% to 55.7%; average measurements were again highest in Group 2 at 44.9%, followed by Group 1 at 42.9% (Table 3, Figure 6). The average bone volume across all scaffold groups was 36.3%.

**Table 3.** Average scaffold and sub-set measurements for bone volume to total volume ratio (BV/TV).

|  | Group 1 | Group 2 | Group 3 | Group 4 |
|---|---|---|---|---|
| Sub-Set 1 | 31.65% | 34.85% | 37.64% | 40.15% |
| Sub-Set 2 | 34.35% | 39.18% | 35.98% | 33.63% |
| Sub-Set 3 | 42.85% | 44.93% | 35.32% | 25.20% |
| Total Average | 36.27% | 39.68% | 36.32% | 32.82% |

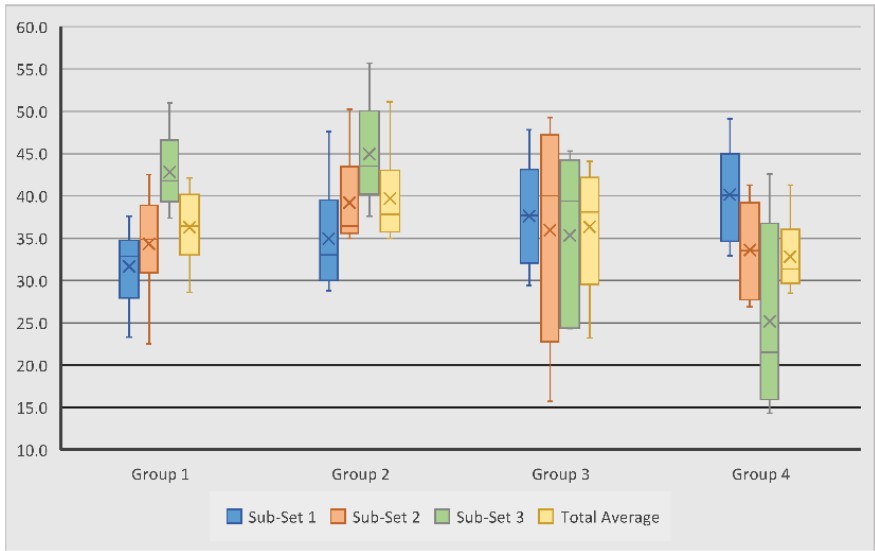

**Figure 6.** Variation of subchondral bone volume to total volume ratio measurements in tested groups and sub-sets.

Scaffold Group 1 and 2 experienced their largest average BV/TV measurements in Sub-set 3. This is consistent with the TbTh measurements for this scaffold group, suggesting that the least bone volume occurred closer to the joint surfaces. On the other hand, in Group 3 and 4, the highest average BV/TV was observed in Sub-set 1, closer to the surface of the joint, with the measurements decreasing in regions further from the surface. This is interesting as the TbTh in Group 3 followed an opposite trend, thus suggesting that the bone volume of the regenerated tissue is greater at the joint surface, despite the thinner trabeculae. However, in Group 4, TbTh did decrease in sub-set regions further from the joint, indicating that the bone volume and trabecular thickness were both greatest closer to the joint surface. Group 4 also experienced the greatest variation between the three sub-sets, with a difference of 15% between Sub-set 1 and 3, suggesting that this scaffold group experienced the least cohesive bone volume regeneration throughout the defect (Figure 7).

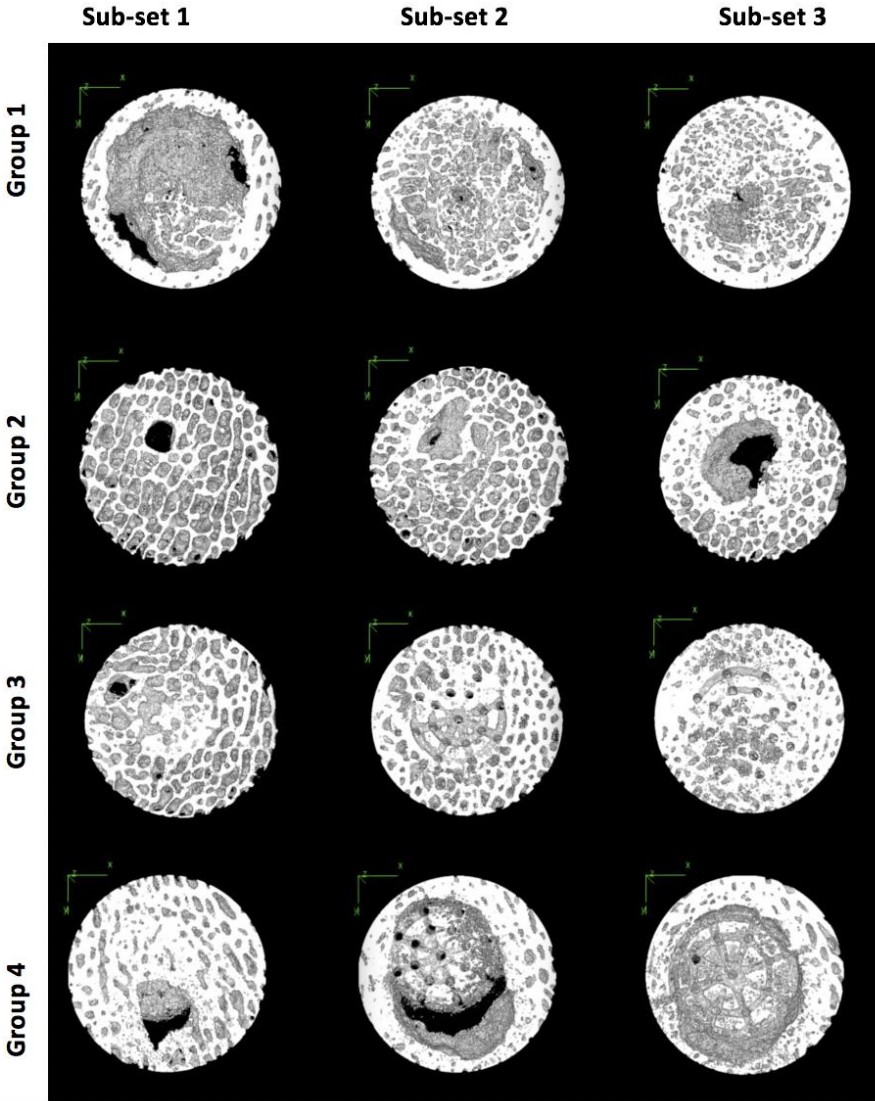

**Figure 7.** Three-dimensional models of samples from each scaffold group and sub-set in the X-Y plane showing bone tissue regeneration.

Statistical analysis showed that the collagen/hydroxyapatite (Group 1 and 2) and Ti-PLA–collagen/PLGA scaffolds (Group 3 and 4), irrespective of BMC addition, generated significantly ($p < 0.05$) different volumes of bone between the three sub-sets (Table 4). No other variables had a significant effect on bone volume regeneration, but other trends within the data were noted. When comparing Group 1 and 3, Group 1 experienced a higher

overall average, but Group 3 contained less variation in the data, suggesting that control scaffolds generated a greater quantity of bone volume, yet the novel scaffold yielded more cohesive volume throughout the defects. Control scaffolds with BMC exhibited a slightly higher average BV/TV than control scaffolds without BMC; however, novel scaffolds without BMC exhibited a slightly higher average value than those with BMC. This again suggests that the addition of BMC did not consistently generate greater osseous regeneration by volume within the tested samples.

**Table 4.** Average bone volume to total volume ratio measurements for all control (HA–Collagen and HA–Collagen + BMC) and all novel (Ti–PLA–collagen/PLGA and Ti–PLA–collagen/PLGA + BMC) scaffold samples by sub-set. These were the only values that exhibited statistically significant ($p > 0.05$) results.

|  | Total Control Average | Total Novel Average |
|---|---|---|
| Sub-Set 1 | 33.30% | 38.90% |
| Sub-Set 2 | 36.77% | 34.81% |
| Sub-Set 3 | 43.89% | 30.26% |

### 3.3. Ti–PLA–Collagen Scaffold Produces Higher Degree of Anisotropy

Degree of anisotropy (DA) was analysed to assess the homogeneity and biomechanical durability of the regenerated bone; this value is always between 0 and 1. The DA for all test groups is summarised in Table 5. It was revealed that scaffold Group 4 experienced the largest overall average DA of 0.33, followed by scaffold Group 3 with 0.31, Group 2 with 0.30, and Group 1 with 0.28. Within Sub-set 1, DA measurements ranged from 0.06 to 0.50, with Group 2 and 4 showing the highest average of 0.34. Measurements in Sub-set 2 and 3 ranged from 0.16 to 0.44 and from 0.13 to 0.46, respectively; Group 4 again showed the highest averages of 0.33 and 0.32 (Figure 8). The average DA across all groups was 0.30.

**Table 5.** Average scaffold and sub-set measurements for the degree of anisotropy.

|  | Group 1 | Group 2 | Group 3 | Group 4 |
|---|---|---|---|---|
| Sub-Set 1 | 0.29 | 0.34 | 0.32 | 0.34 |
| Sub-Set 2 | 0.26 | 0.31 | 0.31 | 0.33 |
| Sub-Set 3 | 0.27 | 0.25 | 0.31 | 0.32 |
| Total Average | 0.28 | 0.30 | 0.31 | 0.33 |

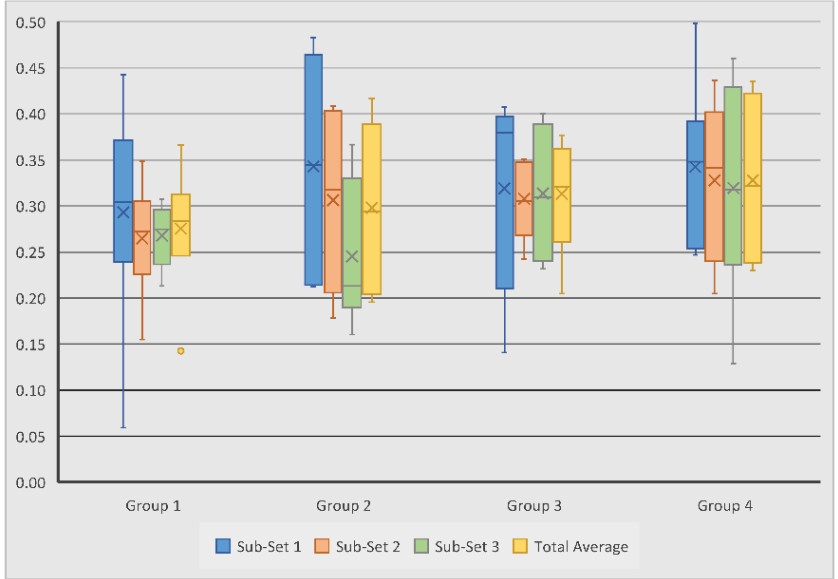

**Figure 8.** Variation of subchondral bone degree of anisotropy measurements in tested groups and sub-sets. No statistically significant differences were observed between the measurements.

In contrast with the other measurements, Sub-set 1 experienced the highest average DA in all four scaffold groups, suggesting that the quality of regenerated bone was consistently better near the joint surface. Within each scaffold group, the difference in DA measurements between the sub-sets never exceeded 0.1, thus suggesting that each scaffold composition generated relatively consistent directionality in bone growth throughout the defect. However, it is important to note that the novel scaffold groups saw less variation than their control group counterparts, indicating that the novel scaffolds did out-perform the control scaffolds in this sense.

When comparing scaffold Groups 1 and 3, Group 3 produced a higher DA average than Group 1; this trend continued in the comparison of Groups 2 and 4, suggesting that novel scaffolds with and without BMC generated higher quality bone than the control scaffolds with and without BMC. Both control and novel scaffolds with BMC also outperformed their non-BMC counterparts, suggesting that the addition of BMC aids in generating subchondral bone tissue which has greater uniformity and durability. The differences between the Groups and the sub-sets were not statistically significant.

*3.4. Macroscopic Evaluation of Scaffold Degradation and Tissue Formation*

After micro-CT evaluations, the samples were cut through the middle to qualitatively evaluate tissue formation and scaffold degradation. As can be seen in Figure 9, the scaffolds in the control groups (Group 1 and Group 2) were entirely degraded and were partly filled with bony tissue. However, subchondral bone oedema and large voids were still observed. In the Ti–PLA–collagen/PLGA scaffolds (Group 3 and Group 4), Ti and PLA matrices can be seen. After six months, 3D-printed PLA was still present in the samples, whereas the collagen sponge was degraded and replaced by cartilage-like tissue.

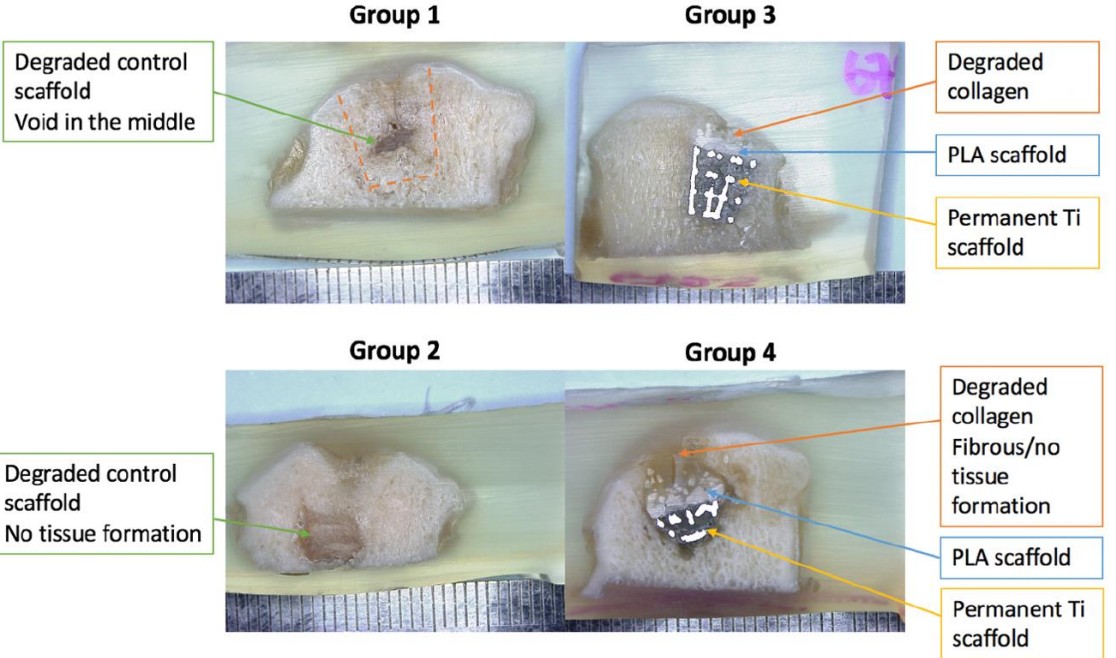

**Figure 9.** Light microscopy images of resin embedded sections to examine the regenerated tissue and scaffold degradation extent.

## 4. Discussion

This study analysed subchondral bone regeneration following osteochondral defects. Using HA–collagen and Ti–PLA–collagen/PLGA scaffolds with and without BMC, data for TbTh, BV/TV, and DA were collected and compared between scaffold groups and sub-set regions. Statistical analysis was used to assess if scaffold composition, the addition of BMC,



and sub-set region contributed to significant differences in subchondral bone regeneration. Finally, microscopic analysis of tissue formation and scaffold degradation was performed.

Trabecular thickness measurements quantify bone growth. As outlined above, none of the TbTh measurements yielded significant results, suggesting that none of the study parameters affected subchondral tissue regeneration significantly. These results are inconsistent with previous researchers who have found that multi-layered scaffolds composed of titanium significantly enhance subchondral bone regeneration [2,36,37]. Given the limited scaffold testing in large-bodied animals and the uncertainty of how trabecular bone will mature both in quantity and quality in response to foreign objects, this is not entirely surprising. Other notable trends were also observed. For example, Sub-set 3 exhibited the largest average TbTh measurements in scaffold Groups 1, 2, and 3, suggesting that at six months post-implantation, the regions furthest from the joint surface experience thicker trabecular growth than those closer to the joint surface. This may be due to the increased biomechanical forces experienced in the subchondral bone closer to the joint surface. When comparing TbTh measurements between scaffold groups, the control and novel scaffolds without BMC exhibited the same average values, suggesting that neither scaffold group was better suited for generating thicker trabeculae. However, the control with BMC did have a higher average than the novel with BMC, indicating that the titanium scaffold utilised in this study did not significantly enhance trabecular thickness as hypothesised.

The BV/TV provides bone volume as a percentage to relay the quantity of growth within the region of interest. Statistically significant results for this measurement were only observed when comparing the effects of the scaffold composition on osseous growth in the sub-set regions, indicating that the regions furthest from the surface experienced significant differences in subchondral bone regeneration between the control and novel scaffolds (irrespective of BMC addition). When visually assessing 3D models of the sub-set, it was initially thought that these results were generated due to the lack of defect repair often observed in that region when compared to Sub-set 1 and 2. While the fact that these measurements were largest in two of the four scaffold groups (Group 1 and Group 2) draws this assumption into question, it is important to cross reference these measurements with the DA to show osseous homogeneity and biomechanical durability of the newly deposited tissue. Sub-set 1 experienced the largest average DA measurements in all four scaffold groups, suggesting that there may not be a direct relationship between the quantity and durability of osseous growth because the anatomical regions with the largest BV/TV in Groups 1 and 2 did not experience sufficient bone quality. This observation is consistent with previous researchers who have reported an inverse correlation between regeneration quantity and quality following osteochondral tissue injuries [38,39]. Additionally, this reinforces the commonly understood biological function, in which the bone closer to the articular surface of a joint experiences larger biomechanical stimuli which results in increased uniformity in the bone deposition. However, Groups 3 and 4 experienced the greatest BV/TV in Sub-set 1, indicating that the novel scaffold both with and without BMC generated greater quality and quantity bone regeneration nearest to the joint surface; this is a promising result as this region experiences the greatest biomechanical forces and requires regenerated bone which can endure them.

Despite the statistical insignificance of the DA measurements, other trends were observed. Scaffolds seeded with BMC experienced higher average DA values in 87.5% of the samples. Though low levels of uniformity (<0.5) were observed in these samples, this trend indicates that the addition of BMC to osteochondral scaffolds may aid in higher quality subchondral bone regeneration, again correlating with results seen in previous studies [38,39]. Ti–PLA–collagen/PLGA scaffolds also generated a larger average DA in 87.5% of the samples, suggesting that the composition of the novel scaffolds examined in this study produce greater biomechanical stability and tissue uniformity than the control scaffolds. These results indicate that, with further examination, the novel scaffold and addition of BMC may lead to sufficient subchondral bone quality.

Macroscopic evaluations of the samples corroborated the findings generated with the micro-CT images regarding tissue growth and unfilled voids in the subchondral compartments. Most of the control scaffolds (Group 1 and Group 2), and the collagenous layers in the Ti–PLA–collagen/PLGA scaffolds (Group 3 and Group 4) were degraded during the implantation period; in some instances, these were replaced with new tissue formation. Group 1 and Group 2 clearly exhibited areas of bone regeneration disproportionately throughout the sub-set regions. The collagen layer in Group 3 and Group 4 scaffolds was replaced with cartilage-like tissue, whilst the other regions experienced osseous growth. The Ti and PLA matrices in the tested scaffolds were still present and identifiable as expected because PLA does not begin to degrade until one and a half years after implantation, and Ti does not degrade once implanted.

While the results of this study suggest that the tested scaffold composition and use of BMC did not generate significant differences in bone regeneration, there may be a few explanations for this occurrence. For example, the present study utilised large-bodied animals, while others studying scaffold treatments for osteochondral tissue defects often use small animals such as rabbits [2,37]. Because larger bodies experience greater load transfers within joints and often larger defects to be healed, the test subjects used may have impacted the results. The use of HA–collagen scaffolds as control samples may have also decreased the observed differences between regeneration in the scaffold composition groups because natural polymers may also aid in osseous growth; this may also explain why studies leaving the defects untreated as a control do see significantly higher levels of growth within their treatments [37]. Finally, it is worth noting that the healing time prior to analysis was longer in the present study than in others [2,36,37]. This may account for the contrasting results observed between previous research and the present study because increased levels of subchondral bone occur earlier in the healing process due to the body's trauma response of depositing excess woven bone. Within the first 2–10 weeks following an injury, bone deposition increases in order to form a callous around the injury; the bone is then reconstructed at an increased rate for as long as a year before an adequate osseous structure is achieved [40]. Given this healing process, it is possible that studies have witnessed greater quantities of bone tissue due to shorter healing periods. It is also possible that the tissue observed in this study was still experiencing rapid remodelling in response to the injury.

## 5. Conclusions

The effect of BMC in combination with osteochondral scaffold (Ti–PLA–collagen scaffold and HA–collagen scaffold) on the bone regeneration and bone microstructure was evaluated in the ovine stifle condyle model. The retrieved tissues were examined by using micro-CT. The microstructural property of the bone surrounding the scaffolds were further analysed using BoneJ software with respect to subchondral trabecular thickness, bone volume to total volume ratio, and degree of anisotropy. The study revealed that both scaffold groups regenerated bone. When observing all samples, the regenerated bone had an average trabecular thickness of 0.29 mm, bone volume ratio of 36.3%, and degree of anisotropy of 0.3; the only statistically significant result was produced when comparing all control and all Ti–PLA–collagen scaffolds with their sub-set regions. The Ti–PLA–collagen scaffolds did not consistently produce a higher quantity of bone in this study; however, they did produce a higher quality of subchondral bone, suggesting that the scaffold composition and addition of BMC enhance subchondral bone homogeneity and biomechanical durability.

**Author Contributions:** Conceptualization: T.F., M.T. and C.L.; Methodology: T.F., M.T. and C.L.; Software: T.F. and M.T.; Validation: T.F. and M.T.; Formal analysis: T.F.; Investigation: T.F., M.T. and C.L.; Resources: T.F., M.T. and C.L.; Data curation: T.F. and M.T.; Writing—original draft preparation: T.F.; Writing—review and editing: M.T.; Visualization: T.F. and M.T.; Supervision: M.T. and C.L.; Project administration: M.T. and C.L.; Funding acquisition: M.T. and C.L. All authors have read and agreed to the published version of the manuscript.

**Funding:** This work was supported by the Versus Arthritis (grant number 21160); and Rosetree Trust (grant number A1184) European Commission via H2020 MSCA RISE programme (BAMOS, grant number 734156); and Innovate UK via Newton Fund (grant number 102872). **Engineering and Physical Science Research Council (EPSRC) (Grant No: EP/T517793/1).**

**Institutional Review Board Statement:** The study was conducted according to the guidelines of the Declaration of Helsinki, and approved by Ethics Committee of the Royal Veterinary College (date of approval 08 August 2014).

**Informed Consent Statement:** Not applicable.

**Data Availability Statement:** The data presented in this study are available on request from the corresponding author.

**Acknowledgments:** The authors would like to thank Tim Arnett for access to micro-CT facilities at UCL and NTL Biologica for providing the bone marrow concentration kits.

**Conflicts of Interest:** The authors declare no conflict of interest.

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
