# Peer review of "Micro-Computed Tomography Analysis of Subchondral Bone Regeneration Using Osteochondral Scaffolds in an Ovine Condyle Model"

_applsci, doi:10.3390/app11030891_

Round 1

Reviewer 1 Report

Author reported the in vivo characterization of two types of osteochondral scaffolds with or without in vitro cell seeding by MicroCT techniques after 6 months implantation. I suggest providing additional tests to improve the quality of the paper and elucidate some important issues.

Authors should add additional characterization of explanted scaffolds to support MicroCT results and to provide qualitative information of regenerated tissues, mainly cellular composition and distribution, blood vessels and new tissue architecture at cellular level.

The study involved only one time point of implantation that is insufficient to clarify if tissue undergoes remodelling from the early implantation period. In fact, as also stated in the discussion section, this aspect did not allow elucidating possible differences in the biological response between the different groups.

The authors should evaluate if scaffolds underwent to degradation in vivo as they were composed of polymers of different composition, such as collagen and plga.

Information regarding scaffolds used are insufficient as Ref 23 is a review articles and the authors did not provide results about morphology, porosity and mechanical properties of scaffolds used.

Reviewer 2 Report

In this research, the authors have assessed if osteochondral scaffolds with addition of bone marrow concentrate (BMC) may enhance tissue regeneration. And they did not observe statistically significant effect of BMC on bone regeneration in osteochondral defects after six months.

The study is described carefully; data is reasonable, and I would like to recommend its publication after minor revisions.

1) There is no information about the care of animals used for the study, e.g:

- what kind of dressing was used,
- whether inflammation has been observed;
- why in one group there are 5 animals not 6/what was the result to exclude one animal from the study

The only information that is given about animals is in lines: 116-122.
There is no information of species and breeder/supplier.

2) a minor remark: Ref 27 is not quoted in the text OR comma should be change to dash in line 107

Author Response

The following information is now added to the manuscript to address the suggested revisions:

Point 1: 

  • Animals were housed in individual pens for 4 days post-surgery and then transferred to group pens for the remainder of the study.
  • Each sheep also had fentanyl patches on pre-operatively until day 3.

  • Post euthanasia, the joints were opened and the defect site and surrounding joint tissues were examined. No skin reaction or inflammation was observed. Tissue reactions ranged from minimal to moderate within the groups.

  • It was not intentional to exclude one animal from the study; unfortunately, the data was lost from that animal and repeat measurements were not possible as the sample was already processed for histology.
  • The breed and supplier is now added: Mules, Royal Veterinary College, UK

Point 2 was an oversight that has now been corrected. 

Round 2

Reviewer 1 Report

Authors provided only minor changes to the original manuscript and there are still important issues that remain unexplained.

Authors justified the absence of cellular results in the paper as these data are outside the main message of this paper that is the "microstructural evaluations of the regenerated bone using Micro-CT". However, microct only allow quantifying new tissue growth, while the assessment of the composition and quality of the regenerated bone is limited with this technique. These are essential aspects of a bone regeneration study, as also evidenced in the manuscript were the authors often stated about “bone regeneration” and “bone quality”.

The correct interpretation of implanted scaffolds response is difficult without the assessment of material degradation. Even if it is true that titanium is non-degradable, all of the other materials and control used in this work probably degraded (in part or completely) during in vivo testing and, their degradation possibly influenced new tissue growth. Furthermore, it is difficult to understand tissue growth in the bone region of osteochondral implants without assessing tissue formation in the entire construct.

Round 3

Reviewer 1 Report

The results and discussion sections have been reasonably improved.